# Magnetite Nanoparticles Coated with PEG 3350-Tween 80: In Vitro Characterization Using Primary Cell Cultures

**DOI:** 10.3390/polym12020300

**Published:** 2020-02-02

**Authors:** Jorge A Roacho-Pérez, Fernando G Ruiz-Hernandez, Christian Chapa-Gonzalez, Herminia G Martínez-Rodríguez, Israel A Flores-Urquizo, Florencia E Pedroza-Montoya, Elsa N Garza-Treviño, Minerva Bautista-Villareal, Perla E García-Casillas, Celia N Sánchez-Domínguez

**Affiliations:** 1Departamento de Bioquímica y Medicina Molecular, Facultad de Medicina, Universidad Autónoma de Nuevo León, Monterrey, Nuevo León 64460, Mexico; jorge.roacho@uacj.mx (J.A.R.-P.); fernandogruiz@outlook.com (F.G.R.-H.); herminiamar@gmail.com (H.G.M.-R.); florencia.estefana@gmail.com (F.E.P.-M.); egarza.nancy@gmail.com (E.N.G.-T.); 2Instituto de Ingeniería y Tecnología, Universidad Autónoma de Ciudad Juárez, Ciudad Juárez, Chihuahua 32310, Mexico; christian.chapa@uacj.mx; 3Facultad de Ciencias Químicas, Universidad Autónoma de Nuevo León, San Nicolas de los Garza, Nuevo León 66455, Mexico; Israel.flores.ur@gmail.com; 4Departamento de Ciencias de los Alimentos, Facultad de Ciencias Biológicas, Universidad Autónoma de Nuevo León, San Nicolas de los Garza, Nuevo León 66455, Mexico; lca.minevillarreal90@gmail.com

**Keywords:** nanoparticles, polyethylene glycol, Tween 80, cytotoxicity, hemotoxicity, primary cell culture, medical applications

## Abstract

Some medical applications of magnetic nanoparticles require direct contact with healthy tissues and blood. If nanoparticles are not designed properly, they can cause several problems, such as cytotoxicity or hemolysis. A strategy for improvement the biological proprieties of magnetic nanoparticles is their functionalization with biocompatible polymers and nonionic surfactants. In this study we compared bare magnetite nanoparticles against magnetite nanoparticles coated with a combination of polyethylene glycol 3350 (PEG 3350) and polysorbate 80 (Tween 80). Physical characteristics of nanoparticles were evaluated. A primary culture of sheep adipose mesenchymal stem cells was developed to measure nanoparticle cytotoxicity. A sample of erythrocytes from a healthy donor was used for the hemolysis assay. Results showed the successful obtention of magnetite nanoparticles coated with PEG 3350-Tween 80, with a spherical shape, average size of 119.2 nm and a zeta potential of +5.61 mV. Interaction with mesenchymal stem cells showed a non-cytotoxic propriety at doses lower than 1000 µg/mL. Interaction with erythrocytes showed a non-hemolytic propriety at doses lower than 100 µg/mL. In vitro information obtained from this work concludes that the use of magnetite nanoparticles coated with PEG 3350-Tween 80 is safe for a biological system at low doses.

## 1. Introduction

Interest in nanotechnology has increased in recent years, looking for its application in different areas, such as agriculture, environmental sciences or even as a solution to various emerging diseases in medicine. Currently, magnetic nanoparticles (MNPs) are one of the most studied nanoparticles in the biomedical field [1,2]. These kinds of nanoparticles are easily manipulated in size, shape and chemical properties. They also have unique physical properties, are biocompatible with the human body, and have a low production cost [3]. MNPs must be designed taking into consideration their physical characteristics to be biocompatible with blood elements and not cause cytotoxicity to other tissues. Nanoparticle size should be from 30 to 150 nm because nanoparticles with smaller sizes are immediately discarded from the blood by the kidneys and large sizes hardly cross biological barriers like epithelia [4]. Due to these characteristics, MNPs have a potential to be applied as an alternative in the diagnosis or treatment of several diseases, such as cancer.

Although MNPs have given good results in preclinical and clinical trials, they can be coated with other materials to improve their proprieties [5,6,7,8]. A coat of polymeric materials is preferred because of their proprieties of biodegradation and biocompatibility [9]. However, a common problem with some polymeric nanoparticles in blood is their high surface charge, which can interact with erythrocytes causing lysis. Polyethylene glycol (PEG) is a nontoxic and non-electrostatic-charge linear polymer [4] that has already been used to coat MNPs. Although there is a common use of PEG in medicine [10,11], the interaction of the PEG molecule with the human body is still being studied [12,13,14]. According to several authors, coating MNPs with PEG improves the MNP blood circulation and their biocompatibility; it reduces their surface charge and maintains the magnetic characteristics of the MNPs [1,5,9]. On the other hand, Tween 80 has been proposed by some authors such as a coat in nanoparticles because of its ability to be targeted into the brain after an intravenous injection. Tween 80 is a biocompatible non-ionic surfactant that binds with the plasma lipoprotein Apo-E, which attaches to Low-Density Lipoprotein (LDL) receptors on the brain micro vascular endothelial cells. This propriety can be used by the nanoparticles in order to delivery substances such as drugs into the brain [15,16]. Tween 80 has also been proposed for the delivery of drugs into cancerous cells [17], has been used as a coating for metallic nanoparticles [18,19], and is widely used in cosmetics, food and pharmaceutics. However, the safety of PEG and Tween 80, avoid the possible harm of nanoparticles is still a challenge for nanomedicine developers. It is a priority to analyze the cytotoxicity of nanoparticles with a focus on the identification of a safe dose range.

Different methodologies for the synthesis of nanoparticles generate nanoparticles with different physical proprieties (size, Z potential, morphology) that leads distinct biological behaviors (cytotoxicity and hemotoxicity). So, for researchers who work in the development of nanoparticles for biomedical applications is important to evaluate the in vitro biological safety of particles to ensure there will be no damage at cellular level. In this study, an in vitro biological characterization was performed with magnetite nanoparticles coated with PEG 3350-Tween 80. Nanoparticle synthesis was developed in order to obtain nanoparticles with physical proprieties, such as size, shape and z-potential, needed for biomedical application. For the evaluation of the nanoparticle interaction with cells, a mesenchymal stem cell (MSC) primary culture from sheep adipose tissue was exposed to different concentrations of magnetic nanoparticles. For cytotoxic assays, primary cell cultures are more sensitive, this is why they are better test subjects than a transformed cell line. The use of undifferentiated cells provides a broader picture of the cytotoxic effects of the nanoparticles on different tissues. Human blood cells from a healthy donor were exposed to the nanoparticles to evaluate hemotoxicity in erythrocytes (hemolysis test) based on the measurement of the lysis of erythrocytes.

## 2. Materials and Methods

This project was reviewed and approved with the identification code BI17-00001 by the ethics in research committee of the School of Medicine and Dr. José Eleuterio Gonzalez University Hospital of the Universidad Autónoma de Nuevo León, approved in September 2017. All animals and human donors were treated according to ethical standards.

### 2.1. Magnetite Nanoparticle Synthesis

Magnetite nanoparticles were synthetized by chemical co-precipitation based on the modified methodology of Chapa-González et al. [20]. We used two precursor solutions: 100 mL of ferric chloride hexahydrate 0.01 M (FeCl_3_·6H_2_O, Thermo Fisher Scientific, Waltham, MA, USA) and 100 mL of ferrous sulfate heptahydrated 0.05 M (FeSO_4_·7H_2_O, Thermo Fisher Scientific, Waltham, MA, USA). Solutions were mixed, agitated and heated until reaching 50 °C. An aliquot of 100 mL of a strong base solution of ammonium hydroxide (NH_4_OH, CTR Scientific, Monterrey, Mexico) was added abruptly to increase the medium pH to produce precipitation of the nanoparticles. The solution was then agitated and heated until it reached 80 °C. The precipitate was washed with distillated water and was isolated by centrifugation three times at 5000 rpm for 10 min until nanoparticles reached a pH of 7. Magnetite nanoparticles were dried at 50 °C for 24 h, and then they were pulverized and saved for functionalization.

### 2.2. Functionalization with PEG 3350-Tween 80

Figure 1 shows the coating process of magnetite nanoparticles. A 25 mg sample of magnetite nanoparticles was dispersed in 25 mL of a 7% Tween 80 (Sigma-Aldrich, St. Louis, MO, USA) solution and 50 mg of PEG 3350 (Thermo Fisher Scientific, Waltham, MA, USA), mixing it with an ultrasonic processor (Fisher Scientific, Pittsburgh, PA, USA); amplitude 80%. The pH was adjusted with ammonium hydroxide until it reached a pH of 9. Nanoparticles were isolated by centrifugation at 3000 rpm for 5 min and washed with distilled water until nanoparticles reached a pH of 7. Nanoparticles were dried at 50 °C for 24 h.

### 2.3. Nanoparticle Characterization

An X-ray diffraction analysis (XDR D2 PHASER; Bruker, Billerica, MA, USA) was performed for the nanoparticles of magnetite. Sample’s XRD pattern was recorded with a LYNXEYE XE-T detector over the 2θ range from 10 to 90° at a scan rate of 0.05°/0.5 s employing Cu-Kα radiation (1.5408 Å). The formed phase was analyzed using a JCPDS PDF card number 01-088-0315 corresponding to magnetite (Fe_3_O_4_). MNPs coat was evaluated by Fourier transform infrared spectroscopy (FTIR NICOLET 6700, Thermo Fisher Scientific, Waltham, MA, USA), to analyze functional groups present in the samples. The measures were carried out in the range from 600 to 4000 cm^−1^ with a 2 cm^−1^ resolution for 130 scanning times.

Nanoparticle morphology and size were examined with scanning electron microscopy (FE-SEM SU5000; HITACHI, Tokyo, Japan). A sample of 3 mg of nanoparticles was dispersed in 3 mL of ultrapure water using an ultrasonic processor. A drop of nanoparticles solution was spread on a carbon tape slide. The sample was dried and subsequently a SEM and an energy-dispersive X-ray spectroscopy (EDS) study were performed. The zeta potential and size distribution in water was measured with dynamic light scattering (DLS) equipment (Zetasizer Nano ZS90, Malvern Instruments, Malvern, UK). The measurements were carried out using a universal dip cell (ZEN 1002, Malvern Instrument, Malvern, UK) at 25 °C. According to literature, nanoparticle samples were dispersed (1000 µg/mL) into a previously filtered ultrapure water immediately before their analysis [21].

### 2.4. Obtention of an MSC Primary Culture

An MSC primary cell culture from sheep adipose tissue was obtained and characterized. The tissue sample was provided by the School of Veterinary Medicine and Zootechnics of the Universidad Autonoma de Nuevo Leon. It came from an independent chirurgical procedure from this study. Subcutaneous fat tissue was obtained from the abdominal region of the sheep. The tissue was transported with supplemented sterile PBS inside a cooler with ice until its processing in a safety cabinet in the laboratory. The supplemented PBS contained PBS pH 7.4 (Thermo Fisher Scientific, Waltham, MA, USA), 0.5% of Gentamicin at 10 mg/mL (Thermo Fisher Scientific, Waltham, MA, USA), 0.5% of Penicillin-Streptomycin at 10 mg/mL (Thermo Fisher Scientific, Waltham, MA, USA) and 0.1% of Fungizone at 250 μg/mL (Thermo Fisher Scientific, Waltham, MA, USA). Fat tissue was washed three times with PBS and three more times with PBS supplemented. With the help of sterile surgical tools, the tissue was fragmented to obtain small portions (mechanical disintegration). Next, a volume of the fragmented tissue was enzymatically disaggregated in two volumes of a 0.2% collagenase I solution. Collagenase I solution contained PBS, 1.2% of Collagenase Type I at 285 U/mg (Thermo Fisher Scientific, Waltham, MA, USA), 0.5% of Gentamicin at 10 mg/mL and 0.1% of Fungizone at 250 μg/mL. The incubation in the collagenase I solution was performed in agitation at 37 °C from 2 to 2.5 h until observing the disintegration of the tissue. The traces of non-disaggregated tissue were eliminated, and individual cells were centrifuged at 5000 rpm for 10 min. The cell button was washed with PBS and finally, resuspended in 1 mL of supplemented MEM medium. The supplemented MEM medium contained Minimum essential medium alpha +GlutaMAX (Thermo Fisher Scientific, Waltham, MA, USA), 10% of SBF (Thermo Fisher Scientific, Waltham, MA, USA), 0.5% of Gentamicin at 10 mg/mL and 0.1% of Fungizone at 250 μg/mL. Cells were cultured for 3-5 days at 37 °C and 5% of CO_2_ until obtained adherent cells with fibroblastic morphology. 

In order to be sure that the cells extracted were MSCs, a characterization by immunocytochemistry was performed. CD-90 goat monoclonal c2441-60 (USBiological, San Antonio, TX, USA) and CD105 mouse monoclonal c2446-55 (USBiological, San Antonio, TX, USA) antibodies were used for immunochemical characterization, using a mouse and rabbit specific HRP/DAB detection system ab64264 (Abcam, Cambridge, UK).

### 2.5. Cytotoxicity: MTT Assay 

To assess whether MNPs could cause damage that affects cell viability, an MTT assay were realized. A total of 5000 MSC were seeded per well in a 96 well microplate and cultured with complemented DMEM. Complemented DMEM contained Dulbecco’s Modified Eagle’s Medium (Thermo Fisher Scientific, Waltham, MA, USA), 10% of SBF and 1% of Penicillin-Streptomycin. After 24 h, cells were exposed with the nanoparticles previously sterilized by UV light exposition. The concentrations evaluated by triplicated were a logarithmic series from 10 to 10,000 µg/mL. Cell death was measured after 24 h of exposition. For MTT assay, cells were incubated with an MTT solution for 4 h. The MTT solution contained MTT (Sigma-Aldrich, St. Louis, MO, USA) at 100 µg/mL in complemented DMEM. After incubation, medium was retired, and formazan was dissolved in 50 µL of isopropanol pH 3. Absorbance was read at 570 nm. 

### 2.6. Hemolysis Test

To measure the impact of nanoparticles in erythrocyte lysis, a quintupled hemolysis test was performed according to Macías-Martínez et al. methodology [22]. Nanoparticles at different concentrations (1, 10, 100 and 1000 µg/mL) were dispersed in Alsever’s solution (dextrose 0.116 M, sodium chloride 0.071 M, sodium citrate 0.027 M and citric acid 0.002 M, pH 6.4). Moreover, a total of 5 mL of blood from a healthy donor was collected in a heparinized-tube (Becton Dickinson, Franklin Lakes, NJ, USA). Erythrocytes were separated by centrifugation (3000 rpm for 4 min) and washed three times using an Alsever’s solution. Nanoparticles and erythrocytes were incubated in a relation of 1:99 erythrocytes: nanoparticles v/v, for 30 min, at 37 °C in agitation at 400 rpm. For positive and negative controls, a suspension of erythrocytes with distilled water and Alsever’s solution, respectively, was used (relation 1:99 v/v). After the incubation, samples were centrifuged at 3000 rpm for 4 min. Hemoglobin released in the supernatant was measured by UV-Vis spectroscopy (NanoDrop, Thermo Fisher Scientific, Waltham, MA, USA) at 415 nm. A positive control with distilled water was considered as 100% hemolysis. Mathematical calculation was required to determine the percentage of hemolysis of problem samples.

### 2.7. Statistical Analysis

For MTT and hemolysis assays a statistical analysis was developed. The cell viability percentages, as well as percentages of hemolysis, were analyzed with an analysis of variance (ANOVA) and a Tukey’s HSD (Honestly Significant Difference) test, with a confidence interval of 95% to determine significant differences between the control group and the test samples.

## 3. Results and Discussion

### 3.1. Magnetite Nanoparticle Synthesis

Magnetite nanoparticles are nanoferrites characterized by having a crystalline structure. Magnetite (Fe_3_O_4_) is an iron mineral that is composed of a combination of oxide Fe^2+^ and Fe^3+^. Magnetite nanoparticle synthesis was performed by the nanoprecipitation method, where the chemical reaction that takes place for the formation of magnetite is represented by the Equation (1) [20]:Fe^2+^ + 2Fe^3+^ + 8OH^−^ → Fe_3_O_4_ + 4H_2_O(1)

The XRD pattern was employed to determine crystal structure and size of the synthesized magnetite nanoparticles. Figure 2a shows the magnetite nanoparticle XRD pattern. It confirms the formation of magnetite in a pure phase because there are no other peaks aside from the magnetite phase. Additionally, according to consulted literature those diffraction peaks correspond to the cubic spinel structure of magnetite (Fe_3_O_4_). Authors report the same diffraction peaks for the magnetite elaborated via co-precipitation [23,24,25,26,27,28]. The determination of the average crystallite size was estimated by the Debye–Scherrer Equation (2)
D = 0.9λ/βcos(θ)(2)
where λ is the wavelength (1.5408 Å), β is the full width at half maximum of the most intense peak (3 1 1) and θ is the diffraction angle of the (3 1 1) peak (35.61 degrees). The sample’s average crystallite size calculated was 9.49 nm. The obtained size is comparable with the references [23,28]. The FTIR spectra of magnetite was recorded and displayed in the Figure 2b, this analysis was done to verify the presence of characteristics bands of magnetite [29]. Bare magnetite nanoparticles show one main band also reported by other authors, corresponding to the presence of Fe–O [23,27,30]. Both, the XRD and the FTIR analyzes, provide information about a reliable methodology for the synthesis of magnetite nanoparticles.

### 3.2. Magnetite Coating with PEG 3350-Tween 80

PEGylation consists in a polymeric conjugation strategy. Due to PEG have two hydroxyl groups at their ends, PEG can adhere to almost any structure [10]. The FTIR spectra of magnetite coated with PEG 3350-Tween 80 was recorded and displayed in the Figure 3. This analysis was done to verify the presence of characteristics bands of magnetite on the sample and PEG 3350-Tween 80 on its surface. The spectrum presents more bands than the bare magnetite spectrum due to different functional groups present in PEG 3350 and in Tween 80. The spectrum shown the vibrational modes corresponding to the vibration modes of magnetite, PEG 3350 and Tween 80. So, was confirmed that magnetite nanoparticles were coated with PEG successfully. The identified vibration modes of PEG and Tween 80 are also described in other reports [27,31,32,33,34,35,36].

### 3.3. Nanoparticle Characterization: Shape

Properties of nanoparticles changes when they are functionalized, so they need to be evaluated [37]. Physical characteristics of the nanoparticles are important to estimate the efficiency of a nanoparticle. Morphology is a propriety that affect in the efficiency of the nanoparticles internalization into cells is nanoparticle geometric morphology [38]. Shape is also correlated with lifetime in blood circulation [39]. Magnetite (Figure 4a) and magnetite nanoparticles coated with PEG 3350-Tween 80 (Figure 4b–d) SEM images are shown. Coated nanoparticles shown a spherical shape. In addition, EDS analysis showed the chemical elements that compound the nanoparticles coated with PEG 3350-Tween80, demonstrating the presence of Fe coming from magnetite inside the samples.

### 3.4. Nanoparticle Characterization: Size

DLS technique was used to analyze the size of the nanoparticles. This technique measures the size of nanoparticles dispersed in an aqueous solution, so if the nanoparticles generates agglomerates, the given size is correlated with the size of the agglomerates [9]. DLS of bare magnetite (Figure 5a) shows single particle average size of 10.51 nm and agglomerates of 527.8 and 4367 nm. The sizes of the bare magnetite single nanoparticle correspond to their average crystallite size given by XRD, so this evidence indicates that magnetite nanoparticles generated are monocrystalline. Although, only the 2.6% of the sample generate agglomerates of almost 4.4 µm, biggest agglomerates can cause several problems if they are administrated by blood into the patient. DLS results of magnetite nanoparticles coated with PEG 3350-Tween 80 (Figure 5b) shows single particle average size of 119.2 nm and agglomerates average size of 785.6 nm. Although, the agglomerates generated by coated nanoparticles are in a bigger proportion than bare nanoparticles, the sizes of the biggest agglomerates of coated nanoparticles (2.7 µm) are approximately 2.3 times smaller than the size of bare nanoparticle agglomerates (6.4 µm). The size of the nanoparticles and their agglomerates affects their elimination and circulation time in the organism. Sizes from 30 to 100 nm could penetrate well into highly permeable tumors [40]. Nanoparticles with a diameter smaller than 5.5 nm results in a rapid urinary excretion [41]. A disadvantage of larger particles is that they are easily recognized by the macrophages from the immune system, while smaller sizes have a longer circulation time in blood [4]. In fact, agglomeration of coated nanoparticles is demonstrated in DLS results, but SEM images shows well defined nanoparticles in the nanometer scale. These agglomerates can be avoided by the application of a mechanical force such as ultrasonication. Magnetite nanoparticles coated with PEG 3350-Tween 80 single nanoparticle have the optimum size because they are large enough for not to be eliminated from the bloodstream due to renal excretion, but, are small enough to cross epithelia and not be quickly eliminated by the macrophages.

### 3.5. Nanoparticle Characterization: Zeta Potential

DLS showed a zeta potential of −3.41 mV for magnetite nanoparticles and a value of +5.61 mV for magnetite nanoparticles coated with PEG 3350-Tween 80. Positively charged nanoparticles are more susceptible to being bound and internalized into cancer cells [42]. However, nanoparticles with positively charged surfaces immobilize serum proteins faster than neutrally charged surfaces. This could be a prejudicial propriety because cationic nanoparticles can be loaded with proteins secreted by the immune cells and be recognized and eliminated from blood circulation [43]. Coated nanoparticles obtained from this work shown a low positively charged surface. Positive charges are also correlated with cytotoxicity and hemolysis. That why authors make an emphasis in the evaluation of the interactions of obtained nanoparticles with cells.

### 3.6. Immunological Characterization of MSC

MSC from adipose tissue are easy to obtain and are available in large quantities, which allows them to be used in different biomedical applications [44]. In this work, the MSC isolated from sheep exhibited fibroblastic morphology with positive expression of CD90 and CD105, which indicates success in the isolation process of MSCs without contamination with other kinds of cells. For the positive control of Anti CD90, a histological section of a sheep brain was used, and the positive control of Anti CD105 was a histological section of a sheep spleen. The positive labeling for the technique was observed through a brown precipitate, which differs from the nuclei in a violet color (Figure 6). CD90 was observed in 88.36% of the cells, while CD105 was observed in the 77.30% of the cells.

### 3.7. Cytotoxicity Test

Since in literature there are several findings of harm nanoparticles that damage circulatory, digestive, endocrine, immune, integumentary, nervous, reproductive, respiratory and urinary systems [45], it is a priority to analyze the cytotoxicity of nanoparticles and the kind of harm they can cause. The 3-(4,5-dimethylthiazol-2-yl)-2,5-diphenyltetrazolium bromide (MTT) is a colorimetric technique that evaluates the metabolic activity of cells, specifically the activity of oxidoreductases enzymes found in the mitochondria. Healthy cells that are exposed to an MTT reagent reduce it into formazan crystals, which are insoluble in water and purple color [46].

The results of this study are shown in Figure 7. At all tested concentrations (from 10 to 10,000 µg/mL), bare magnetite nanoparticles showed no statistical difference (*p* = 0.05) in comparison with the negative control. In the magnetite nanoparticles coated with PEG 3350-Tween 80, only concentrations from 10 to 1000 µg/mL showed no significant difference against the negative control (*p* = 0.05); so at these doses, there is no damage in the cell metabolism and the nanoparticles are not considered a dangerous substance. On the other hand, statistical analyses had shown that at concentrations of 10,000 µg/mL, magnetite nanoparticles coated with PEG 3350-Tween 80 displayed a significant difference with respect to the negative control (*p* = 0.05). Thus, it is not safe to use coated nanoparticles at concentrations of 10,000 µg/mL and higher.

### 3.8. Interactions with Erythrocytes Membranes

Determine if nanoparticles can cause erythrocyte lysis in an ex vivo model is important because when hemolysis occurs in vivo it can cause anemia, jaundice and other pathological conditions. In addition, the hemoglobin released can have a toxic effect on the renal and vascular system. The hemolysis test measures the lysis of the red blood cells exposed to an environmental agent. This lysis produces the release of the intracellular content of the erythrocyte due to the rupture of its membrane. The measured released molecule was the hemoglobin, which is a predominant protein in erythrocytes. According to Standard Practice for Assessment of Hemolytic Properties of Materials, ASTM F756-17, the hemolytic activity of materials is classified in three types: non-hemolytic materials (0%–2% of hemolysis), low hemolytic materials (2%–5% of hemolysis) and high hemolytic materials (higher than 5% of hemolysis) [22]. As is shown in Figure 8 magnetite nanoparticles shown a non-hemolytic behavior at all concentrations tested (from 1 to 1000 µg/mL). On the other hand, magnetite nanoparticles coated with PEG 3350-Tween 80 are safe only at concentrations lower than 100 µg/mL. There was a difference between bare magnetite nanoparticles, which were non-hemolytic at the concentration of 1000 µg/mL, against nanoparticles coated with PEG 3350-Tween 80, which were high hemolytic at the same concentration. This grade of hemotoxicity can be attributed to the positive charges in the coated nanoparticle surface [4].

Although there are many advantages of the use of PEG 3350 and Tween 80, such as prolonging the circulation time in the body [12], the correct formulations and concentrations in the use of the coating must be measured and tested because excess of coating can cause several problems at the cellular level. In this work, we found that coating the MNPs did not improve their biological in vitro properties, as observed in the hemolysis and MTT test, but we find that the use of the coat eliminates the presence of agglomerates bigger than 2.7 µm. Some authors findings [47,48,49] show that PEGylated magnetite nanoparticles are safe to use, but we have to take into consideration that the preparation of the nanoparticle influences in the result of the cytotoxicity tests, as well as the nature of the cell lines or primary cultures employed, the election of the test performed, the PEGylated magnetite doses administrated, the time of exposure and the molecular weight of the PEG employed. Liu et al. in 2017 presented a complete cytotoxic evaluation of different molecular weight PEG molecules. They found that different molecular weight molecules correspond to different cytotoxic behaviors. They also corroborated that at higher doses PEG is more cytotoxic. Liu et al. contributed the PEG cytotoxicity to the reactive oxygen species (ROS) generation, which is generated by PEG derivates [50]. In this study we presented an analysis of the PEG degradation, which showed the disappearance of bonds from the PEG polymer generating derivatives, which correlates also with the formation of ROS. There is no extensive exploration of the formation of ROS and how they affect PEG biological proprieties; a new challenge is opened for researchers who wish to add PEG to their nanostructures for the purpose of improving physical properties of nanoparticles. Another option would be to compare PEG cytotoxicity with different polymers that have given good results in the functionalization of MNPs such as silica [51], chitosan [52] or polyethylenimine [9].

## 4. Conclusions

In this study, we were able to develop magnetite nanoparticles coated with PEG 3350-Tween 80, as the XRD, EDS and FTIR analyses showed. The SEM and DLS studies showed spherical morphologies and nanometric sizes. The main point of this study was the comparison of the cytotoxicity of bare magnetite nanoparticles compared to magnetite nanoparticles coated with PEG 3350-Tween 80. At low doses of magnetite nanoparticles coated with PEG 3350-Tween 80 are safe to use. The authors recommend the use of primary cell cultures over cell lines to test the cytotoxicity of nanoparticles, due to their sensitivity and similarity to an in vivo environment. Now that authors have developed a non-toxic nanoparticle with the possibility of been targeted to the brain, new in vitro and in vivo testes will be required in order to develop an application.

## Figures and Tables

**Figure 1 polymers-12-00300-f001:**
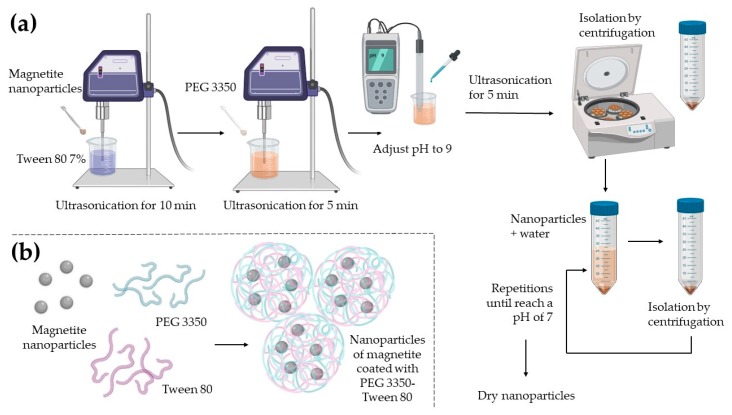
A scheme cartoon of the coating process of nanoparticles with PEG 3350 and Tween 80. (**a**) A complete diagram of the coating process methodology. (**b**) The interaction of PEG 3350 and Tween 80 with the magnetite nanoparticles is possible because of physical forces, creating a polymer network around the magnetite nanoparticles.

**Figure 2 polymers-12-00300-f002:**
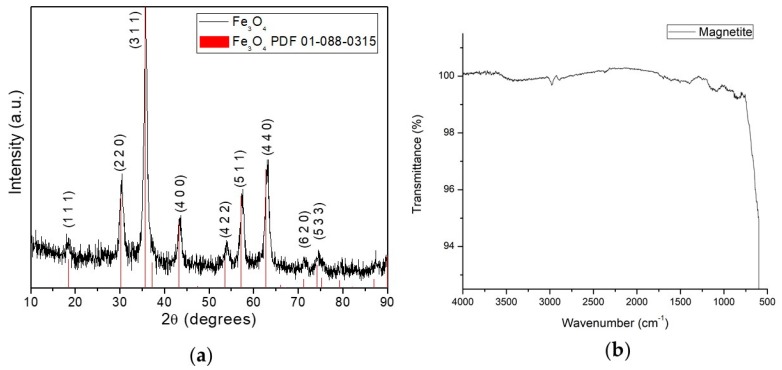
Magnetic nanoparticles (MNPs) synthesis characterization. (**a**) XRD pattern from magnetite nanoparticles exhibits nine diffraction peaks observed at 2θ value 18.29, 30.13, 35.61, 43.26, 53.84, 57.32, 63.13, 71.15 and 74.17, which match with the reference PDF 01-088-0315 card. Those peaks are indexed to the (1 1 1), (2 2 0), (3 1 1), (4 0 0), (4 2 2), (5 1 1), (4 4 0), (6 2 0) and (5 3 1) planes respectively; (**b**) FTIR spectra of bare magnetite. Magnetite shows the first band around 600 cm^−1^. It is related to Fe–O stretching in the octahedral site of the crystal structure of magnetite.

**Figure 3 polymers-12-00300-f003:**
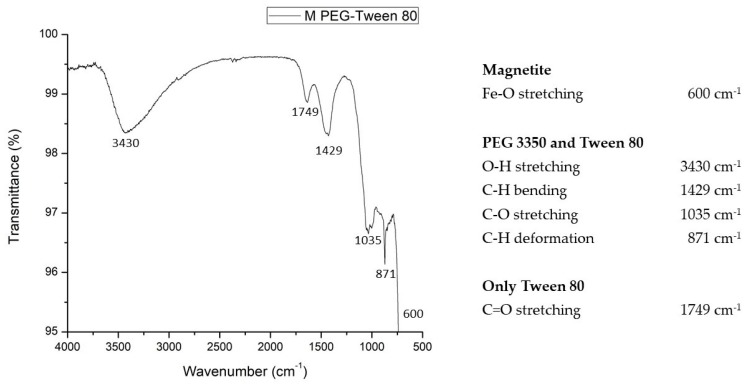
FTIR spectra of magnetite coated with PEG 3350-Tween 80. Obtained bands correspond to bonds found in PEG 3350 as well as in Tween 80: deformation (871 cm^−1^) and bending (1429 cm^−1^) vibration of C–H; C–O stretching (1035 cm^−1^) and the O-H stretching (3430 cm^−1^). Samples present the Fe–O stretching (600 cm^−1^) coming from the magnetite and the C=O stretching (1749 cm^−1^) coming from the Tween 80.

**Figure 4 polymers-12-00300-f004:**
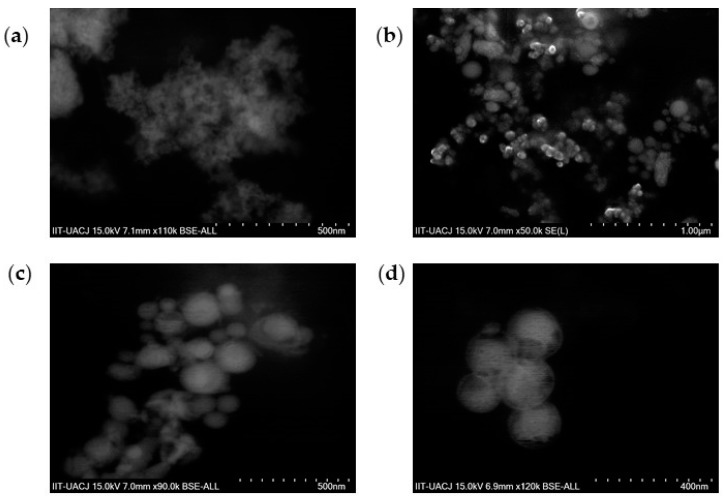
SEM images. (**a**) Magnetite nanoparticles and (**b**–**d**) magnetite nanoparticles coated with PEG 3350-Tween 80 at different magnifications. All the samples showed spherical shapes. In bare magnetite nanoparticles are evidence of agglomeration, while in magnetite nanoparticles coated with PEG 3350-Tween 80 the separation between one nanoparticle and other becomes more evident.

**Figure 5 polymers-12-00300-f005:**
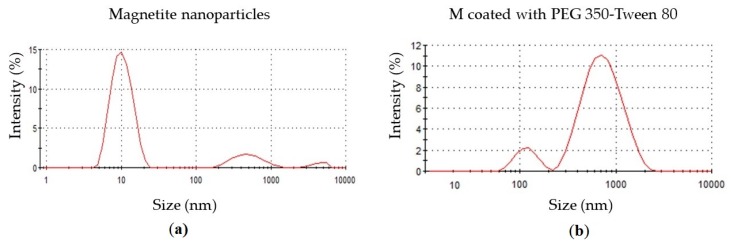
Dynamic light scattering (DLS) sizes results. (**a**) Bare magnetite nanoparticles size distribution. With an average size of 10.51 nm 83.8% of the nanoparticles do not generate agglomerates. The 13.6% of the nanoparticles generates agglomerates of 527.8 nm and the other 2.6% generates agglomerates of 4367 nm. (**b**) Magnetite nanoparticles coated with PEG 3350-Tween 80 size distribution. With an average size of 119.2 nm 10.3% of the nanoparticles do not generate agglomerates. The 89.7% of the nanoparticles generates agglomerates of 785.6 nm.

**Figure 6 polymers-12-00300-f006:**
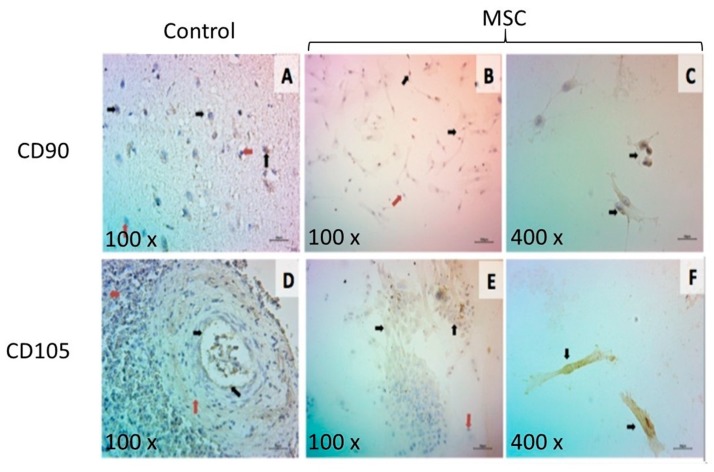
Immunocytochemistry characterization of a mesenchymal stem cell (MSC). (**A**) Ovine brain tissue positive for CD90; (**B**,**C**) MSC marked with CD90; (**D**) ovine spleen tissue positive for CD105 and (**E**,**F**) MSC marked with CD105.

**Figure 7 polymers-12-00300-f007:**
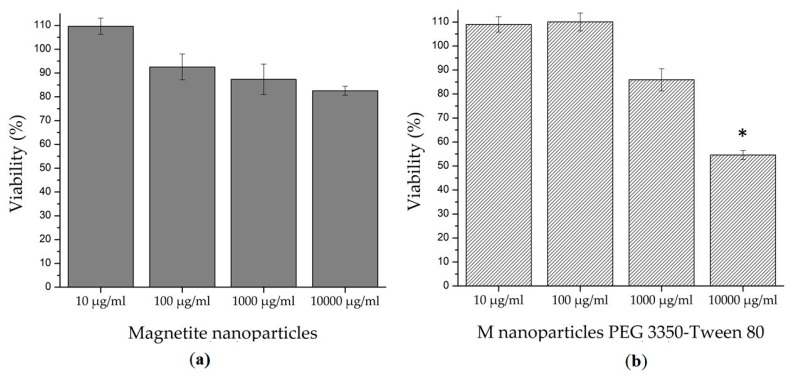
MTT assay results. Both nanoparticles tested, (**a**) bare magnetite nanoparticles and (**b**) magnetite nanoparticles coated with PEG 3350-Tween 80, shown a non-cytotoxic behavior at concentrations from 10 to 1000 µg/mL. At concentrations of 10,000 µg/mL, only magnetite nanoparticles coated with PEG 3350-Tween 80 show a cytotoxic effect with a significant difference* in comparison to the negative control.

**Figure 8 polymers-12-00300-f008:**
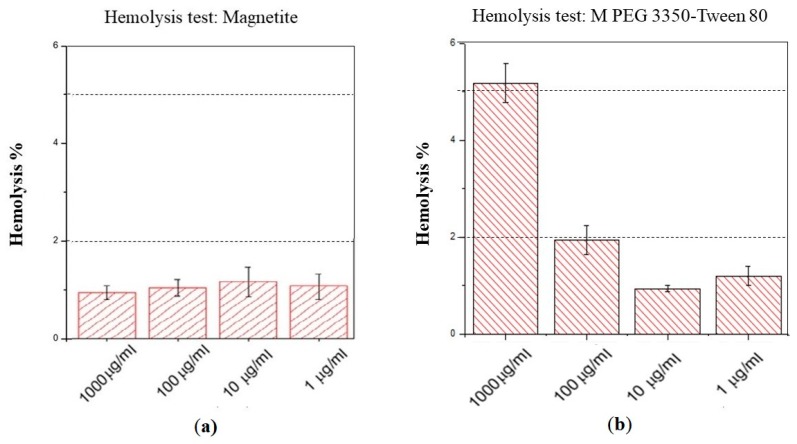
Hemolysis test results. (**a**) Bare magnetite nanoparticles and (**b**) magnetite nanoparticles coated with PEG 3350-Tween 80. Results show that both nanoparticles are considered safe from 1 to 100 µg/mL. Bare magnetite nanoparticles are safe at concentrations of 1000 µg/mL, while magnetite nanoparticles coated with PEG 3350-Tween 80 have shown hemotoxicity at that concentration.

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
