# Peer review of "Magnetite Nanoparticles Coated with PEG 3350-Tween 80: In Vitro Characterization Using Primary Cell Cultures"

_polymers, 2020, doi:10.3390/polym12020300_

Round 1

Reviewer 1 Report

No more comments

Author Response

The authors thank to the reviewer for your revision. 

Minor english language and style changes have been done based also in editors comments. 

Reviewer 2 Report

The novelty of the manuscript is slightly improved compared with last version, and some solid data are added.

A scheme cartoon should be provided to show the coating process of nanoparticles with PEG. 

The authors should explain why the data in Figure 6a are different that in the last version. In addition, why the date at the concentration of 1 μg/ml was removed?

Round 2

Reviewer 2 Report

Accept after minor revise.

The authors are required to double check the details. For example, "tween 80" in Figure 1 should be "Tween 80".

Author Response

The authors thank to the reviewer for your revision.

Minor english language details have been done based in your comments. All changes have been highlighted using the "Track Changes" function of Microsoft Word for make it easily visible to the editors and reviewers.

This manuscript is a resubmission of an earlier submission. The following is a list of the peer review reports and author responses from that submission.

Round 1

Reviewer 1 Report

Overall, the paper describes a standard synthesis of magnetite nanoparticles and their postulated functionalization with PEG as well as biocompatibility tests using RBC lysis, MTT metabolic activity and morphological evaluation. The synthesis of PEGylated magnetic particles has been a very active field in, at least, the last decade (as also pointed out by the authors; line 39) and so numerous studies exist on toxicological evaluations of nanoparticles, especially of various iron oxides. It is difficult for this reviewer to see what new knowledge the current study brings to this field.

Furthermore, this reviewer feels compelled to point out some misinterpretations of the data in the paper:

The PEGylation was performed by dispersing 25 mg nanoparticles in 25 mL (water) with 500 mg TritonX100 and 50 mg PEG 3350. Then adjusting the pH. Then washing the particles with successive centrifugation and redispersing with water. This will at max lead to some degree of adsorbtion of TritonX100(major) or PEG(minor) on the surface and will not provide a stable anchorage of the PEG to the NP surface especially when further diluted in the subsequent tests performed. PEG is very hydrophilic and without any form of anchorage it would be expected to be washed out during this procedure. The identified vibration modes detected by FTIR might as well (more likely) be from residual adsorbed TritonX100.
NP surface layers can be hard to characterize properly and many times specialized equipment such as X-ray Photoelectron Spectroscopy will be required. This could help the interpretation of the exact make-up of the coating layer in this paper. If sufficient signal can be achieved, perhaps the coating can be entirely eluted from the nanoparticles and NMR can be obtained to measure the contribution from PEG and TritonX100, respectively. The term “oxygen bond” is not suitable here (oxygen bonds could perhaps form between two oxygen atoms, creating the di-oxygen molecule) or perhaps for peroxides. In this case, by loss of water, the resulting bond would (hypothetically) be an ether bond. This is, however, very unlikely to be formed here and requires radically different reaction conditions! The images on figure 2, a and b do not demonstrate individual particles. This reviewer questions the method of measuring particle size by image analysis of images of this quality level. Triton X-100 is a surfactant and toxic to (otherwise relatively robust) HeLa cells from 0.15 mM (doi: 10.1073/pnas.1011614107). Here, Triton X100 is used as a dispersant prior to PEG modification in a concentration of 30 mM and then washed. Due to the high amounts used, some amount could still remain adsorbed, depending on the washing procedures, and this could perhaps be the reason why the PEGylated particles display higher toxicity as observed in the MTT and RBC lysis study. TritonX100 is also used as dispersant before DLS measurements (actually electrophoretic mobility) for measuring NP zeta potential. The presence of nonionic surfactants will enshroud any charges on the NP surface and make zetapotential closer to neutral. In effect, this will hamper the quality and usefulness of the data obtained from the zetasizer. No real differences in zetapotential were observed. Line 289: PEG (poly(ethylene glycol)) is NOT a polyester, it is a polyether! It is not biodegradable as such. This is the whole issue with PEGs for parenteral use. Small molecular weight PEGs will, however, excrete due to being below the renal threshold. Reference 39 in your manuscript deals primarily with polyESTERS.

Other specific concerns:

Line 46: “Sharp corners can cause damage to vessels..”. This statement does not make sense and belongs in a macro-setting. The authors are encouraged to think on the actual dimensions of nanoparticles and blood vessels.

Line 73: What is the reason that the ethics statement is in the introduction?

Line 238: What is on the axis’ in figure 1, c and d? Why is N increased for the “PEGylated” nanoparticles? Why is O not higher for PEGylated nanoparticles?

Line 311: Figure 4, c and d. Please use the same scale on the axis for better readability.

Reviewer 2 Report

Herein, the authors prepared and studied PEGylated magnetite nanoparticles in primary cell cultures. In brief, the article is interesting and valuable for researchers working in biomedical fields. Therefore, it can be reconsidered for publication after a revision process. 

1) The text should be polished by a native speaker 

2) FTIR study. Obviously new bands appeared after the PEGylation process such as peaks at 1076, 2852, and 2921 cm-1. The authors should state to which bands they belong. 

3) SEM image is blurry and the size cannot be determined well. Instead, a TEM study is suggested. 

4) It is highly suggested to study the zeta potential stability of PEGylated magnetite nanoparticles at different pH which is close to biological range. 

5) Figure 4. Again, the TEM study is better here to see the size variation. 

6) It is unclear why PEGylated magnetite nanoparticles degrade faster compared to bare nanoparticles? 

7) Highly relevant studies can be added such as: DOI: 10.1007/s12668-017-0447-6  and DOI: 10.1039/C3NR02188B

Reviewer 3 Report

Lots of papers have been reported on polymer grafted magnetic nanoparticles for biological applications. This manuscript is just a repeat work on preparation of PEGylated magnetic nanoparticles. Without any new performance for application, I can’t find any novelty from this manuscript.  

In addition, some important date is not persuasive. The resolution of SEM images is too low. Higher resolution TEM images should be provided.

Take Figure 2a as an example, it is unbelievable the authors could realize they are spherical nanoparticles and obtained the size distribution via ImageJ.

The authors said PEGylated magnetite nanoparticles are homogeneous according to their date obtained by ImageJ (Figure 2d). However, it is obvious that the particles are not uniform, with size ranges from 20 to 80 nm (Figure 2b). I don’t believe the size distribution been shown in Figure 2d is correct.